# Real-Life Diagnostic Performance of the Hypersensitivity Pneumonitis Guidelines: A Multicenter Cohort Study

**DOI:** 10.3390/diagnostics13142335

**Published:** 2023-07-11

**Authors:** Ophir Freund, Yitzhac Hadad, Tamar Shalmon, Ori Wand, Sonia Schneer, Tal Moshe Perluk, Eyal Kleinhendler, Tzlil Hershko, Boaz Tiran, Galit Aviram, Evgeni Gershman, Yochai Adir, David Shitrit, Amir Bar-Shai, Avraham Unterman

**Affiliations:** 1Center of Excellence for Interstitial Lung Diseases, Tel Aviv Medical Center, Tel Aviv University, Tel Aviv-Yafo 6801298, Israel; ophir068@gmail.com (O.F.);; 2Institute of Pulmonary Medicine, Tel Aviv Medical Center, Tel Aviv University, Tel Aviv-Yafo 6801298, Israel; talmp@tlvmc.gov.il (T.M.P.);; 3Department of Radiology, Tel Aviv Medical Center, Tel Aviv University, Tel Aviv-Yafo 6801298, Israel; 4Division of Pulmonary Medicine, Barzilai University Medical Center, Ashkelon 7830604, Israel; 5Pulmonary Division, Lady Davis Carmel Medical Center, Faculty of Medicine, The Technion Institute of Technology, Haifa 3200003, Israel; 6Pulmonary Department, Meir Medical Center, Kfar Saba 4428164, Israel; davids3@clalit.org.il

**Keywords:** interstitial lung disease, hypersensitivity pneumonitis, bronchoalveolar lavage, diagnosis, accuracy

## Abstract

Hypersensitivity pneumonitis (HP) is a heterogeneous interstitial lung disease (ILD) that may be difficult to confidently diagnose. Recently, the 2020 ATS/JRS/ALAT HP diagnostic guidelines were published, yet data validating their performance in real-life settings are scarce. We aimed to assess the diagnostic performance of the HP guidelines compared to the gold-standard multidisciplinary discussion (MDD). For this purpose, we included consecutive ILD patients that underwent diagnostic bronchoscopy between 2017 and 2020 in three large medical centers. Four diagnostic factors (antigen exposure history, chest computed tomography pattern, bronchoalveolar lavage lymphocyte count, and histology results) were used to assign guidelines-based HP diagnostic confidence levels for each patient. A sensitivity analysis was performed, with MDD diagnosis as the reference standard. Overall, 213 ILD patients were included, 45 (21%) with an MDD diagnosis of HP. The guidelines’ moderate (≥70%) confidence threshold produced optimal performance with 73% sensitivity for HP, 89% specificity, and a J-index of 0.62. The area under the receiver operating characteristic curve (AUC) for a correct guidelines-based diagnosis was 0.86. The guidelines had better performance for non-fibrotic than fibrotic HP (AUC 0.92 vs. 0.82). All diagnostic factors, except bronchoalveolar lavage lymphocyte count, were independent predictors for MDD diagnosis of HP in a multivariate analysis. In conclusion, the HP guidelines exhibited a good diagnostic performance compared to MDD diagnosis in real-life setting.

## 1. Introduction

Hypersensitivity pneumonitis (HP) is an immune-mediated interstitial lung disease (ILD) following inhalational exposure to an inciting antigen [1]. The disease is classified as fibrotic or non-fibrotic based on high-resolution computed tomography (HRCT) findings [2,3]. Since antigen identification and avoidance is associated with improved survival in patients with HP [4,5], correct diagnosis is important for early and proper intervention. HP has a wide heterogeneity in presentation and findings, with no widely accepted gold-standard diagnostic test or algorithm [6,7,8]. Four main factors should be considered in the diagnostic workup of HP: history of antigen exposure, HRCT, bronchoalveolar lavage (BAL), and lung biopsy [5,9,10,11,12]. The multidisciplinary discussion (MDD) diagnostic approach is the current gold standard for ILD diagnosis [1,2,3,13]. However, implementing this approach is limited by poor agreement between experts about the importance of diagnostic factors and lack of ILD experts and resources in smaller health care facilities [14,15]. These limitations highlight the need for widely accepted and validated diagnostic criteria.

A promising step towards consensus is the recent HP diagnostic guidelines of the American Thoracic Society, Japanese Respiratory Society, and Asociacion Latinoamericana del Torax (ATS/JRS/ALAT), issued in 2020 [3]. These guidelines define HP as “an immune-mediated disease that manifests as ILD in susceptible individuals after exposure to an identified or unidentified factor”. They specify in detail each of the four diagnostic factors, including the sources of antigens known to be associated with HP, criteria for chest HRCT and histopathologic findings associated with fibrotic and non-fibrotic HP, and the cutoff for BAL lymphocyte counts compatible with HP. The guidelines then tabulate the various combinations of the four diagnostic factors, assigning five confidence levels for HP diagnosis ranging from “HP not excluded” to “definite diagnosis of HP”. The guidelines also offer an algorithm for the diagnosis of HP based on their recommendations, including an exposure assessment and chest HRCT as the initial steps in all cases and BAL lymphocyte cellular analysis (recommendation in non-fibrotic HP and only a suggestion in fibrotic HP). Lastly, the guidelines specify considerations for histopathological sampling in non-fibrotic and fibrotic HP.

While the guidelines are detailed and based on prior research, real-life data are urgently needed to validate their utility. Our aim was to assess the diagnostic performance of the 2020 ATS/JRS/ALAT HP guidelines’ criteria in real-life settings, compared to the gold-standard MDD diagnosis.

## 2. Materials and Methods

### 2.1. Study Population and Design

This is a retrospective, observational, multicenter cohort study of consecutive ILD patients that underwent diagnostic bronchoscopy between January 2017 and December 2020. Data were obtained from ILD registries of three large medical centers in Israel. All included patients were adults (above 18 years old) with an available HRCT scan that was conducted in the six months before or after the bronchoscopy and with access to their clinical data. Patients with a non-ILD diagnosis were excluded. We also excluded patients with a high-confidence diagnosis of sarcoidosis, which is usually clinically and radiologically straight-forward to diagnose, thus avoiding a biased elevated diagnostic accuracy of the ATS/JRS/ALAT guidelines [3,16]. The study was approved by the Tel Aviv Sourasky Medical Center review board (21-0456-TLV) and conducted per the Declaration of Helsinki. Due to the nature of this retrospective study and the preserved anonymity of patients, a waiver of informed consent was given by the TASMC institutional review board. The study is reported according to the STrengthening the Reporting of OBservational studies in Epidemiology (STROBE) checklist.

To evaluate the performance of the ATS/JRS/ALAT guidelines, we examined the concordance between the “gold-standard” MDD diagnosis of HP (MDD-HP) and the confidence level of HP diagnosis determined by the guidelines.

### 2.2. Data Collection and MDD Diagnosis

Demographic and clinical data were obtained from registries and based on pulmonologist visits and any other relevant encounters such as hospitalizations or clinic visits. Clinical data included occupational and environmental exposures, family history, co-morbid diseases, medications, pulmonary functions tests, and treatment. We re-reviewed the records of pulmonologist visits to identify any additional antigen exposure history. HRCT scans, BAL results, and histopathology were accessed from institutional systems. BAL results included a manual differential cell count and the CD4/CD8 T-cell ratio (by flow cytometry) [17]. Lung histopathology was obtained by bronchoscopy (trans-bronchial forceps biopsy or cryobiopsy), according to the clinical judgement of the pulmonologist.

We defined the MDD diagnosis as our reference standard for ILD diagnosis [3,13]. MDD diagnosis was determined for all ILDs according to accepted standards and performed separately in each center [18]. The multidisciplinary teams did not use the HP guidelines’ criteria for diagnosis, nor were they aware of the HP guidelines-based diagnostic confidence assigned to each patient.

### 2.3. The ATS/JRS/ALAT Guidelines Diagnostic Criteria

The ATS/JRS/ALAT guidelines’ criteria rely on four main diagnostic factors to determine the confidence level of HP diagnosis: history of antigen exposure, HRCT scan, BAL, and histopathological results [3]. We re-evaluated each factor for every patient according to these guidelines to calculate the diagnostic confidence level. Positive history of antigen exposure was defined as a history of exposure to any antigen mentioned in the guidelines with time relevance to symptoms or imaging findings [3]. All patients were treated by ILD specialists and their initial evaluation in the clinic included a thorough environmental and occupational exposure history. Serum-specific immunoglobulin G assays for HP were not performed, as they are not available in Israel. Assessment of HRCT scans was conducted by an expert chest radiologist who was blinded to the clinical data and the MDD diagnosis. We categorized all HRCT scans as “fibrotic” or “non-fibrotic” and then analyzed them for radiologic features of HP patterns according to the guidelines’ criteria. Each HRCT scan was considered typical, compatible, or indeterminate for HP. The guidelines do not include indeterminate HRCT features for non-fibrotic ILD; thus, patients without any typical or compatible findings were categorized as “other diagnosis”. As the guidelines do not recommend a specific BAL lymphocytosis threshold, a BAL lymphocyte count above 20% was used in our primary analysis with additional exploratory analysis for the 30% threshold [3,19]. Histopathological results were categorized according to the guidelines’ criteria as HP, probable HP, indeterminate for HP, other diagnosis, or a non-diagnostic specimen. The term trans-bronchial biopsy refers to both trans-bronchial forceps biopsy and trans-bronchial cryobiopsy, unless otherwise stated.

Based on the results of the four main factors above and the guidelines’ criteria, we calculated the HP diagnostic confidence level for each patient: definite HP (>90% diagnostic likelihood), high confidence (80–89%), moderate confidence (70–79%), low confidence (51–69%), and HP not excluded (≤50%).

### 2.4. Statistical Analysis

Continuous variables were reported as mean (±standard deviation) or median (interquartile range) for normally and non-normally distributed variables. Normality was evaluated using the Kolmogorov–Smirnov test. Chi-square and independent *t*-tests were used to compare categorical and continuous variables, respectively. The Mann–Whitney U-test was used to compare between non-normally distributed continuous variables. Independent predictors for MDD-HP diagnosis were assessed using a multivariate logistic regression analysis. Sensitivity, specificity, accuracy, positive predictive value (PPV), negative predictive value (NPV), and J-index (Youden’s index) were derived from two-by-two tables with 95% confidence intervals (95% CI) when relevant. Sensitivity analysis was performed for the guidelines’ HP diagnostic confidence levels, each time with a different confidence level as a threshold for diagnosis. We calculated the probability of making a guidelines-based HP diagnosis for every ILD diagnosis in our cohort. The discriminative ability is the ability of the algorithm to correctly classify patients and reflects its overall predictive performance. We calculated the discriminative ability of the guidelines by measuring the area under the receiver operating characteristic (ROC) curve. The classification thresholds in the ROC curve were the four diagnostic confidence levels of the guidelines. In general, an area under the ROC curve >0.75 is considered to be consistent with a good discriminant ability [20]. The significance level was set at 0.05. Data were analyzed using SPSS version 28 (IBM Corp., Armonk, NY, USA).

## 3. Results

A total of 213 patients were included in the study cohort, 45 (21%) with a MDD-HP diagnosis and 168 (79%) with MDD-nonHP diagnoses. All patients had available HRCT scans, and all underwent bronchoscopy, 189 (89%) with trans-bronchial biopsy and 159 (75%) with BAL. Overall, 122 patients (57%) had a fibrotic ILD and 91 (43%) had a non-fibrotic disease. The most prevalent non-HP ILD diagnoses were idiopathic pulmonary fibrosis (IPF, 30 patients,) and connective tissue-related ILD (CTD-ILD, 30 patients), as presented in Figure 1.

### 3.1. Diagnostic Performance of the Guidelines

The performance of the guidelines for each of their diagnostic confidence levels is shown in Table 1. The guidelines-based HP diagnosis with a threshold level of moderate confidence (≥70%) had a sensitivity of 73% (95% CI: 58–85%), specificity of 89% (95% CI: 84–94%), and the highest J-index of 0.62. Compared with the moderate confidence level as a threshold for HP diagnosis, the low confidence level showed a drop in specificity to 68% (95% CI: 60–75%), and the high confidence level showed a drop in sensitivity to 51% (95% CI: 36–66%). Overall, negative predictive values were high (84% to 95%) with variable positive predictive values (42% to 81%). The guidelines’ criteria had a higher sensitivity for non-fibrotic HP compared to fibrotic HP (84% vs. 65%) with similar specificity (89–90% for both). The area under the ROC curve (AUC) for a correct diagnosis by the guidelines was 0.86 (Figure 2, blue line).

In 17 patients, the guidelines-based HP diagnosis (moderate confidence and above) was different than the MDD diagnosis. The discordant MDD diagnoses were idiopathic nonspecific interstitial pneumonia (iNSIP, *n* = 4, 21% of patients with iNSIP), IPF (*n* = 4, 13%), organizing pneumonia (OP, *n* = 3, 13%), CTD-ILD (*n* = 2, 7%), granulomatous and lymphocytic interstitial lung diseases (GLILD, *n* = 1), unclassifiable ILD (*n* = 1), drug-related ILD (*n* = 1), and smoking-related ILD (*n* = 1). Trans-bronchial biopsy was performed in all of these patients; one was non-diagnostic (no alveolar tissue), twelve provided a definite diagnosis consistent with the MDD diagnosis, three did not provide a definite diagnosis but findings were compatible with the MDD diagnosis, and only in one case was the biopsy compatible with HP (although HRCT was not; therefore, the MDD diagnosis was unclassifiable ILD). A total of 15 of the 17 mentioned patients (88%) had a confident MDD diagnosis or provisional diagnosis with high confidence (i.e., ≥70% confidence diagnosis according to Ryerson et al.) [21].

In a sub-analysis among patients that underwent trans-bronchial biopsy (*n* = 189, 105 forceps biopsy and 84 cryobiopsy), the area under the ROC curve for a correct diagnosis by the guidelines was 0.87 (Figure 3, blue line). When the biopsy results of these patients were not factored in the guidelines’ diagnostic criteria, the area under the curve was 0.84 (Figure 3, orange line).

### 3.2. Patient Characteristics and Predictors for MDD-HP Diagnosis

The cohort’s baseline characteristics are presented in Table 2, with a comparison between patients with and without an MDD-HP diagnosis. The two groups did not differ with regards to age, smoking status, comorbidities, or lung function tests. The only statistically significant difference was a higher female representation in the MDD-HP group (62% vs. 42%).

The majority of patients with MDD-HP diagnosis were females (62%) and 44% were past or current smokers. Twenty-nine (64%) had a known antigen exposure (Table 3), which included: mold (*n* = 9), birds (mainly pigeons and parrots, *n* = 16), farming (*n* = 3), and spray paint (*n* = 1). The active treatments of patients with MDD-HP varied, with 18 patients (40%) receiving prednisone, 11 (24%) receiving prednisone and additional steroid-sparing immunosuppressive (IS) therapy, 4 (9%) receiving steroid-sparing IS therapy alone, and 12 (27%) without IS treatment. Six patients (13%) received anti-fibrotic drugs.

Table 3 presents the results of the four main diagnostic factors between patients with and without an MDD-HP diagnosis. Patients with MDD-HP diagnosis had higher rates of all four diagnostic factors, including (1) HP-related antigen exposure, (2) typical or compatible HRCT pattern, (3) above 20% lymphocytes in BAL, and (4) histopathological results of probable or typical HP (*p* < 0.001 for all).

In the multivariate logistic regression analysis (Table 4), a BAL lymphocyte count of above 20% was the guidelines’ only diagnostic factor that did not remain an independent predictor for an MDD-HP diagnosis (adjusted OR: 2.23, 95% CI: 0.75–6.63). BAL lymphocyte count remained a non-significant factor when the threshold of above 30% was used in the multivariate analysis (*p* = 0.397). Female sex was also a non-significant predictor in this analysis.

## 4. Discussion

We evaluated the performance of the ATS/JRS/ALAT guidelines’ criteria for HP diagnosis in comparison to MDD diagnosis among 213 ILD patients that underwent bronchoscopy in three large medical centers in Israel. To the best of our knowledge, this is the first study to test the guidelines’ performance in a real-world setting, while including both fibrotic and non-fibrotic HP and a variety of other ILDs.

The guideline diagnostic criteria had an overall high specificity, NPV, and diagnostic accuracy. Its ability to distinguish between HP and non-HP patients was demonstrated by an area under the ROC curve of 0.86 (Figure 2), which is considered as an excellent discriminating ability [22]. A guideline diagnostic confidence ≥70% (moderate) demonstrated a sensitivity of 73% and specificity of 89%, which seems to be the best-balanced combination of the two. This confidence level also had the highest J-index (0.62) and maximal point on the ROC curve, and we believe it should be the threshold used for the guidelines criteria. Buendia-Roldan et al. [23] showed that the ATS/JRS/ALAT guidelines were restrictive in determining a confident diagnosis, which was also reflected in our study by the drop in sensitivity (51%) when a higher confidence level was used.

Diagnostic uncertainty is more substantial in fibrotic than in non-fibrotic HP, given the overlapping radiological and histopathological features with other fibrotic ILDs [24]. A similar trend was found in our study, demonstrated by higher sensitivity for non-fibrotic HP of 84% compared with 65% for fibrotic HP. This was also evident by the lower AUC for fibrotic HP compared with that of non-fibrotic disease (0.82 vs. 0.92, Figure 2). Takei et al. [25] showed the guidelines had a sensitivity of 74% for patients with fibrotic HP, consistent with our findings. However, their calculated specificity was lower than ours (70% vs. 90%), possibly secondary to the larger percentage of unclassifiable ILDs in their study (26% vs. 7%).

The guidelines do not indicate a specific BAL lymphocyte threshold to distinguish HP from other ILDs. A systematic review by Patolia et al., conducted in the context of development of the HP guidelines, showed a higher BAL lymphocyte count among HP patients without a threshold that resulted in an acceptable sensitivity and specificity [19]. Their findings were supported by others who also demonstrated a good diagnostic yield for high BAL lymphocyte counts [26,27]. The optimal BAL lymphocyte threshold for HP diagnosis is still unknown, with limited change in performance between the different BAL lymphocyte cutoffs [3,25,28]. In our cohort, a BAL lymphocyte count of above 20% was a strong predictor for MDD-HP diagnosis in univariate analysis, while after adjusting for the other diagnostic factors in a multivariate model, it was not a significant independent predictor. Although an interesting finding, we cannot rule out sample size issues.

Whether trans-bronchial biopsy should be a part of the initial evaluation of every patient with suspected HP is still not determined [3]. We showed that when the moderate confidence level is used for diagnosis, the histological results had only a mild impact on the final diagnosis (Figure 3). This means that when a patient has a relevant exposure with a typical HRCT pattern, the diagnostic impact of a biopsy is questionable. However, based on our multivariate analysis, typical or probable HP histological findings were independent predictors for MDD-HP. We believe that this finding reflects the important role of a biopsy in more complicated cases when diagnosis remains undetermined after the initial evaluation.

This study is not without limitations. Selection bias is possible by including only patients that underwent bronchoscopy. However, we believe that our cohort represents a relevant patient population for optimal guidelines implementation, in which bronchoscopy is an important part. In addition, the trans-bronchial biopsy type (forceps or cryobiopsy) was selected by the treating physician which could affect its yield. However, this also represents a real-world scenario. Data were collected retrospectively, affecting mainly the exposure history which was not standardized. Other diagnostic factors (i.e., HRCT, BAL lymphocytes, and histology) are less likely to be affected by the retrospective study design.

## 5. Conclusions

The ATS/JRS/ALAT HP guidelines’ diagnostic criteria had good discriminating ability and accuracy compared with the MDD diagnosis for both fibrotic and non-fibrotic HP, with optimal performance at the moderate confidence level. Based on our results, clinicians can use these guidelines to consolidate their HP diagnosis and determine the diagnostic level of confidence. While not a substitute for MDD, in places with limited resources or expertise, these guidelines may be used to formulate a working diagnosis of HP, start appropriate treatment, and advocate for antigen avoidance. Future studies should investigate the ability of these guidelines to shorten the time for HP diagnosis and initiation of appropriate interventions, especially when applied to non-expert settings where an MDD is not readily available.

## Figures and Tables

**Figure 1 diagnostics-13-02335-f001:**
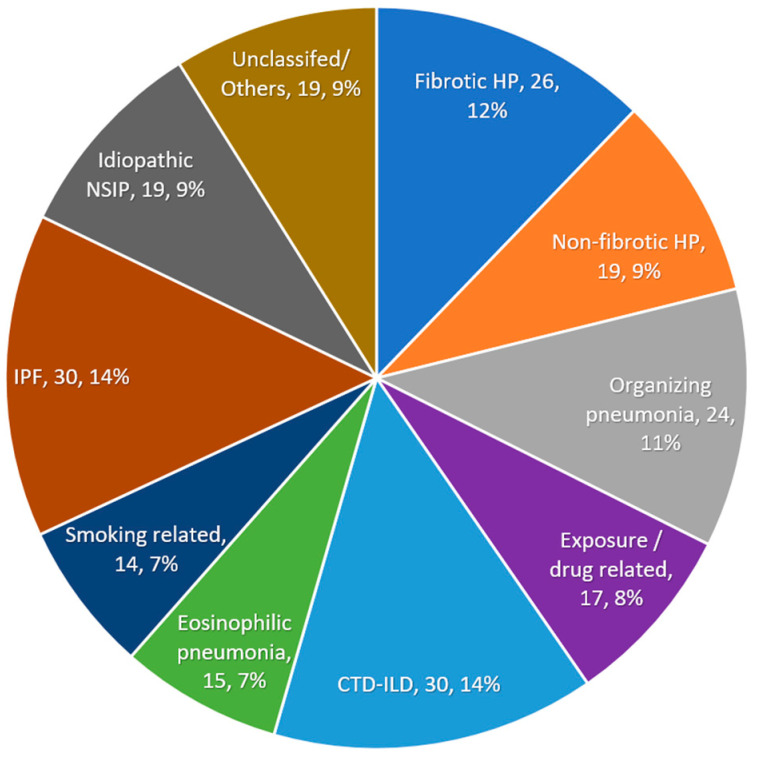
Distribution of multidisciplinary discussion diagnoses (*n* = 213). Abbreviations: HP, hypersensitivity pneumonitis; CTD, connective tissue disease; IPF, idiopathic pulmonary fibrosis; NSIP, nonspecific interstitial pneumonia; ILD, interstitial lung disease.

**Figure 2 diagnostics-13-02335-f002:**
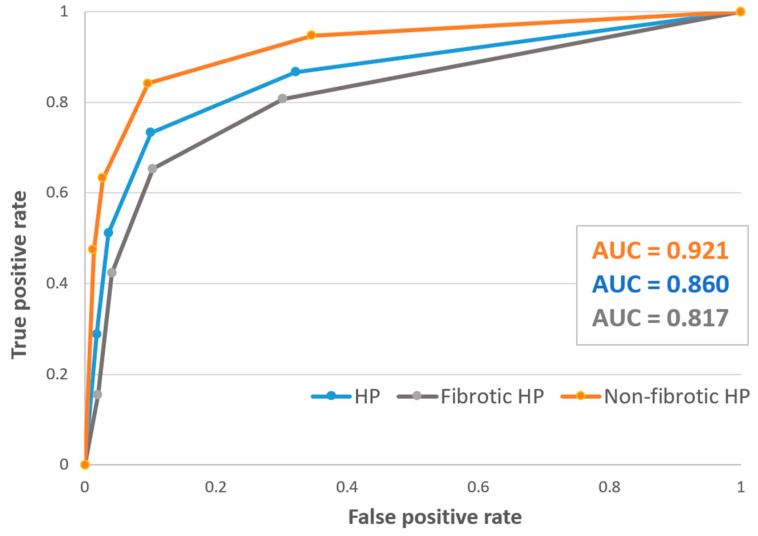
Receiver operating characteristic (ROC) curve of the HP guidelines’ diagnostic performance for HP (blue line), fibrotic HP (grey line), and non-fibrotic HP (orange line). Abbreviations: HP, hypersensitivity pneumonitis; AUC, area under the ROC curve.

**Figure 3 diagnostics-13-02335-f003:**
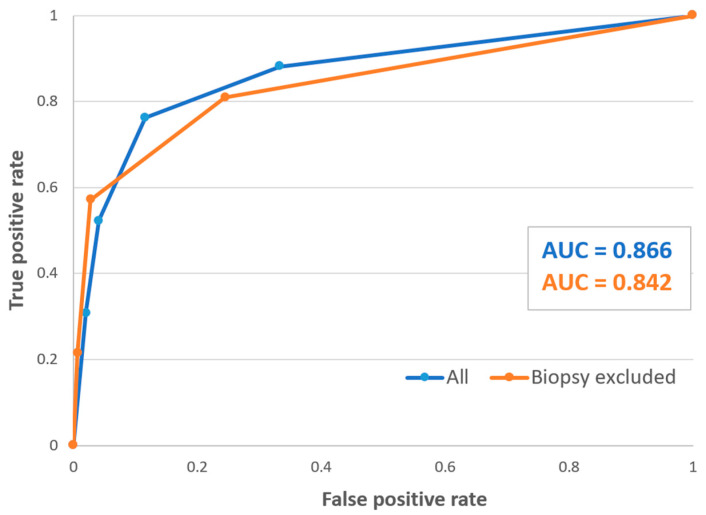
Receiver operating characteristic (ROC) curve of the HP guidelines’ diagnostic performance for all patients that underwent TBB (*n* = 189), with (blue line) and without (orange line) the TBB results factored in. Abbreviations: HP, hypersensitivity pneumonitis; TBB, trans-bronchial biopsy; AUC, area under the ROC curve.

**Table 1 diagnostics-13-02335-t001:** Performance of the ATS/JRS/ALAT guidelines for each diagnostic confidence level.

Guidelines Diagnostic Confidence *	Sensitivity % (95% CI)	Specificity † % (95% CI)	PPV † % (95% CI)	NPV † % (95% CI)	Accuracy † % (95% CI)	J-Index †
Low	87 (73–95)	68 (60–75)	42 (36–48)	95 (90–98)	72 (65–78)	0.55
Moderate						
Overall	73 (58–85)	89 (84–94)	66 (55–76)	93 (86–95)	86 (81–91)	0.62
Fibrotic HP	65 (44–83)	90 (82–95)	63 (47–77)	90 (85–94)	84 (77–90)	0.55
Non-fibrotic HP	84 (60–96)	89 (79–95)	67 (50–80)	96 (88–98)	88 (80–94)	0.73
High	51 (36–66)	96 (92–98)	79 (62–90)	88 (85–91)	86 (82–91)	0.47
Definite	29 (16–44)	98 (95–100)	81 (56–94)	84 (81–86)	84 (78–88)	0.27

* Confidence levels calculated using the guidelines criteria (low (51–69%), moderate (70–79%), high (80–89%), and definite (≥90%) diagnostic confidence). Each confidence level in this analysis represents a threshold for HP diagnosis by the guidelines. † Variables were calculated in relation to MDD diagnosis of hypersensitivity pneumonitis.

**Table 2 diagnostics-13-02335-t002:** Baseline characteristics of the study cohort, compared between patients with and without an MDD diagnosis of HP.

Variable	MDD-HP * *n* = 45 (%)	MDD-NonHP * *n* = 168 (%)	*p* Value
Female sex	28 (62)	71 (42)	0.017
Age, years (SD)	62 (±13)	62 (±14)	0.939
Current or former smokers	20 (44)	85 (50)	0.464
Smoking pack years, median (IQR)	0 (0–20)	5 (0–28)	0.651
Hypertension	17 (38)	59 (35)	0.741
Diabetes mellitus	9 (20)	36 (21)	0.835
Additional (non-ILD) lung disease	4 (9)	18 (11)	0.781
Gastroesophageal reflux disease	12 (27)	33 (20)	0.305
Fibrotic lung disease	26 (58)	96 (57)	0.939
Lung function tests:			
FEV1%—predicted (SD)	75 (±17)	78 (±16)	0.436
FVC%—predicted (SD)	75 (±15)	79 (±15)	0.332
FEV1/FVC (SD)	0.80 (±0.1)	0.82 (±0.1)	0.496
TLC%—predicted (SD)	83 (±20)	83 (±22)	0.889
DLCO%—predicted (SD)	54 (±18)	56 (±17)	0.601

Abbreviations: MDD, multidisciplinary discussion; HP, hypersensitivity pneumonitis; FEV, forced expiratory volume; FVC, forced vital capacity; TLC, total lung capacity; DLCO, diffusing capacity for carbon monoxide. Results are presented as percentages with standard deviations (SD) or as medians with interquartile range (IQR) for normally distributed and non-normally distributed continuous variables, respectively. * MDD diagnosis of HP or interstitial lung disease other than HP.

**Table 3 diagnostics-13-02335-t003:** Results of the four main diagnostic factors for HP between patients with and without an MDD diagnosis of HP.

	MDD-HP * *n* = 45 (%)	MDD-NonHP * *n* = 168 (%)	*p* Value
(1) HP-related antigen exposure	29 (64)	35 (21)	<0.001
(2) Typical or compatible HRCT pattern for HP	36 (80)	65 (39)	<0.001
(3) Bronchoalveolar lavage: †			
Lymphocytes above 20%	19 (49)	24 (20)	<0.001
Lymphocytes above 30%	12 (31)	15 (12.5)	0.008
Lymphocytes % (IQR)	18 (11–37)	8 (4–17)	<0.001
Macrophages % (IQR)	43 (17–64)	39 (16–63)	0.923
Neutrophils % (IQR)	18 (6–36)	27 (10–49)	0.083
Eosinophils % (IQR)	2.2 (1–4)	2.4 (0.5–5)	0.838
CD4/CD8 ratio (IQR)	1.7 (1–3)	1.2 (0.5–2)	0.163
(4) Trans-bronchial biopsy **:			
Probable or typical for HP	18 (40)	17 (10)	<0.001
Indeterminate	12 (27)	68 (40)	0.089
Non-indicative or missing	10 (22)	51 (30)	0.284
Other diagnosis	5 (11)	32 (19)	0.212

Abbreviations: MDD, multidisciplinary discussion; HP, hypersensitivity pneumonitis; HRCT, high-resolution computed tomography. Results are presented as medians with interquartile ranges (IQR). * MDD diagnosis of HP or interstitial lung disease other than HP. ** Including both trans-bronchial forceps biopsy (*n* = 105) and trans-bronchial cryobiopsy (*n* = 84). † Median and interquartile range were used because of non-normally distributed results. Only patients that underwent bronchoalveolar lavage were included in this analysis (*n* = 159, 75%).

**Table 4 diagnostics-13-02335-t004:** Multivariate regression analysis of predictors for MDD diagnosis of HP *.

Variable	AOR	95% CI	*p*
Female sex	1.90	0.66–5.46	0.231
Antigen exposure	11.01	3.79–28.03	<0.001
Typical or compatible HRCT pattern for HP †	5.14	1.68–13.69	0.004
BAL lymphocytes > 20%	2.23	0.75–6.63	0.148
Probable or typical HP histological findings †	5.99	1.65–18.70	0.006

Abbreviations: MDD, multidisciplinary discussion; HP, hypersensitivity pneumonitis; AOR, adjusted odds ratio; HRCT, high-resolution computed tomography; BAL, bronchoalveolar lavage. * The analysis only includes patients that had BAL and trans-bronchial biopsy results (*n* = 135). † The HRCT and histological findings are divided based on the ATS/JRS/ALAT HP guidelines.

## Data Availability

All data generated or analyzed during this study are included in this article. Due to ethical and privacy concerns, the primary dataset cannot be made openly available. The study was conducted retrospectively and according to the regulations of our institution’s review board, such data could be openly shared. Requests for the dataset supporting our results can be made via helsinki@tlvmc.gov.il and will be fulfilled by the first author after approval.

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
