# Peer review of "Real-Life Diagnostic Performance of the Hypersensitivity Pneumonitis Guidelines: A Multicenter Cohort Study"

_diagnostics, 2023, doi:10.3390/diagnostics13142335_

Round 1
Reviewer 1 Report
Thank you for giving me the opportunity to revise this practically useful article entitled "Real-life Diagnostic Performance of the Hypersensitivity Guidelines: a Multicenter Cohort Study". The article deals with the problem of the clinical utility of recently proposed diagnostic guidelines for HP. The methodology is well described and the results are clear.
I would recommend enriching the introduction by adding some more information about how the guidelines define HP and what are the main recommendations included there.
In the Tables 2 and 3, if the CI or IQR or SD are reported, it should be indicated in the legend.
The finding that BAL lymphocytosis was a single non-significant factor predicting MDD-HP diagnosis needs more discussion. There are many publications indicating opposite findings (e.g. 1. Adderley N., Humphreys C.J., Barnes H., Ley B., Premji Z.A., Johannson K.A. Bronchoalveolar lavage fluid lymphocytosis in chronic hypersensitivity pneumonitis: A systematic review and meta-analysis. Eur. Respir. J. 2020;56:2000206. doi: 10.1183/13993003.00206-2020, 2. Patolia S., Tamae Kakazu M., Chami H.A., Chua A., Diaz-Mendoza J., Duggal A., Jenkins A.R., Knight S.L., Raghu G., Wilson K.C. Bronchoalveolar Lavage Lymphocytes in the Diagnosis of Hypersensitivity Pneumonitis among Patients with Interstitial Lung Disease. Ann. Am. Thorac. Soc. 2020;17:1455–1467. doi: 10.1513/AnnalsATS.202005-420OC 3. Sobiecka M, Szturmowicz M, Lewandowska KB et al. Bronchoalveolar Lavage Cell Count and Lymphocytosis Are the Important Discriminators between Fibrotic Hypersensitivity Pneumonitis and Idiopathic Pulmonary Fibrosis. Diagnostics (Basel). 2023 Mar 1;13(5):935. doi: 10.3390/diagnostics13050935.) and this point of view should also be included.
Author Response
Reviewer 1 report:
Thank you for giving me the opportunity to revise this practically useful article entitled "Real-life Diagnostic Performance of the Hypersensitivity Guidelines: a Multicenter Cohort Study". The article deals with the problem of the clinical utility of recently proposed diagnostic guidelines for HP. The methodology is well described and the results are clear.
We appreciate the reviewer’s in-depth review and insightful comments, and are grateful for the positive feedback.
I would recommend enriching the introduction by adding some more information about how the guidelines define HP and what are the main recommendations included there.
We thank you for your suggestions for improving the introduction. We elaborated on the definition of HP by the guidelines, the diagnostic criteria and diagnostic algorithm offered in them, and their recommendations regarding the evaluation of patients suspected to have HP (Introduction, lines 52-65).
In the Tables 2 and 3, if the CI or IQR or SD are reported, it should be indicated in the legend.
Thank you for this correction. The use of CI, IQR, or SD has been included in the legend and in each relevant raw in Tables 2 and 3.
The finding that BAL lymphocytosis was a single non-significant factor predicting MDD-HP diagnosis needs more discussion. There are many publications indicating opposite findings (e.g. 1. Adderley N., Humphreys C.J., Barnes H., Ley B., Premji Z.A., Johannson K.A. Bronchoalveolar lavage fluid lymphocytosis in chronic hypersensitivity pneumonitis: A systematic review and meta-analysis. Eur. Respir. J. 2020;56:2000206. doi: 10.1183/13993003.00206-2020, 2. Patolia S., Tamae Kakazu M., Chami H.A., Chua A., Diaz-Mendoza J., Duggal A., Jenkins A.R., Knight S.L., Raghu G., Wilson K.C. Bronchoalveolar Lavage Lymphocytes in the Diagnosis of Hypersensitivity Pneumonitis among Patients with Interstitial Lung Disease. Ann. Am. Thorac. Soc. 2020;17:1455–1467. doi: 10.1513/AnnalsATS.202005-420OC 3. Sobiecka M, Szturmowicz M, Lewandowska KB et al. Bronchoalveolar Lavage Cell Count and Lymphocytosis Are the Important Discriminators between Fibrotic Hypersensitivity Pneumonitis and Idiopathic Pulmonary Fibrosis. Diagnostics (Basel). 2023 Mar 1;13(5):935. doi: 10.3390/diagnostics13050935.) and this point of view should also be included.
We agree that there are opposing findings by different publications indicating the significant role of BAL lymphocytosis in the diagnosis of HP and we thank the reviewer for the valuable examples. We want to highlight that BAL lymphocytosis was found to be a significant predictor for a final diagnosis of HP in our univariate model (Table 2, p<0.001). Only after adjusting for the other diagnostic factors in our multivariate model, it was found to be non-significant with the diagnosis of HP. We do not underestimate the value of BAL cell analysis in the evaluation of HP, as there is a large body of evidence for its diagnostic yield, as mentioned by the reviewer. We accordingly addressed this issue in our discussion with the use of the mentioned studies (lines 326-335).
Reviewer 2 Report
In this paper, the authors presented data from a real-life study comparing diagnostic performance for HP of 2020 ATS/JRS/ALAT HP guidelines with the conclusion of a multidisciplinary discussion (as the gold standard). The authors demonstrated that the 2020 ATS/JRS/ALAT HP guidelines had good diagnostic performance, especially for non-fibrotic HP.
The study is interesting, well written and clear enough, and I have no major concerns.
However, a one point worth mentioning:
- the characteristics of patients with HP are not described in sufficient details, and despite thаt some parameters are presented in tables 2 and 3, it is necessary to provide data on the causes/triggers of HP, symptoms, therapy.
Author Response
Reviewer 2 report:
In this paper, the authors presented data from a real-life study comparing diagnostic performance for HP of 2020 ATS/JRS/ALAT HP guidelines with the conclusion of a multidisciplinary discussion (as the gold standard). The authors demonstrated that the 2020 ATS/JRS/ALAT HP guidelines had good diagnostic performance, especially for non-fibrotic HP.
The study is interesting, well written and clear enough, and I have no major concerns.
However, a one point worth mentioning:
- the characteristics of patients with HP are not described in sufficient details, and despite thаt some parameters are presented in tables 2 and 3, it is necessary to provide data on the causes/triggers of HP, symptoms, therapy.
We would like to thank the reviewer for his efforts in reviewing our manuscript and the positive remarks. The characteristics of the patients with an MDD diagnosis of HP are interesting and important to support the validity of our results. We accordingly provided the data on the causes/triggers, and therapy for patients with HP diagnosis (lines 227-233).